# Comparison of two IGRA assays exploring cell-mediated immunity against CMV, BKV, and EBV in kidney transplant patients

Aurélie Truffot,[1] Soizic Daniel,[2] Lucie Fusillier,[1] Baptiste Nicolas-Truphemus,[1] Heidi Primard,[1] Julien Lupo,[1] Thomas Jouve,[3] Gregory Hansen,[4] Marie Deroussent,[2] Franck Berthier,[5] Raphaële Germi[1]

**ABSTRACT** Monitoring cellular immune responses (CMIs) appears to be a new biomarker in managing immunocompromised patients, especially kidney transplant recipients. This study compared the performance of two interferon-gamma (IFN-γ)-based assays—enzyme-linked immunosorbent spot assay (ELISpot) and VIDAS-IGRA—for detecting T-cell responses against cytomegalovirus (CMV), Epstein-Barr virus (EBV), and polyomavirus BK (BKV). Twenty-eight blood samples from kidney transplant recipients were analyzed using both assays. CMV-specific CMI was assessed using overlapping peptide pools from pp65 and IE-1 proteins (JPT Peptide Technologies GmbH) and a CMV lysate (bioMérieux). All patients were monitored for opportunistic viral replication through qPCR for up to 6 months after blood sampling. For CMV-CMI, the agreement between both assays was 91.7%, 100%, and 80% after stimulation by pp65 peptide pool, pp65 + IE-1 peptide pools, and CMV lysate, respectively. Positive responses were consistent with CMV serology, especially for the VIDAS-IGRA assay and after activation by JPT's PepMix. For EBV-specific responses, positive assay agreement was 92.3%, whereas the positive agreement with BKV was 80.7%. For EBV and BKV, responses were generally weaker than those for CMV (positive response if >10 spots/250,000 PBMC for EBV and >10 spots/350,000 PBMC for BKV in ELISpot or level of IFN-γ >0.08 IU/mL in VIDAS). This study highlights the importance of standardizing interferon-gamma release assays (IGRA). While VIDAS-IGRA RUO offers an open, automated, and standardized solution, responses were in agreement with ELISpot-T assay and even more when using JPT's PepMIX. However, the cellular response to BKV is weaker, and no link with viral replication has been found. This raises the question of the value of these IGRA assays in assessing the cellular response to BKV and the need to optimize these tests. A longitudinal study to observe the evolution of the cellular response could be interesting.

**IMPORTANCE** This original article, a proof of concept, is the first study comparing a new interferon-gamma release assay, performed on whole blood and based on an enzyme-linked immunofluorescent assay detection technique, with a homemade enzyme-linked immunosorbent spot assay using activation by the same peptides. These results were also analyzed in relation to serology results. The inclusion of these tests in clinical guidelines for monitoring cytomegalovirus, Epstein-Barr virus, and polyomavirus BK infections requires studies to evaluate their robustness and compare them.

**KEYWORDS** cell-mediated immunity, interferon gamma release assay, kidney transplantation, opportunistic infections

In immunocompromised patients, particularly solid organ transplant (SOT) recipients, the reactivation of latent viruses, such as cytomegalovirus (CMV), Epstein-Barr virus

Address correspondence to Aurélie Truffot, atruffot@chu-grenoble.fr.

Dr. Berthier and Dr. Daniel are employed by the bioMérieux company.

(EBV), and polyomavirus BK (BKV), is a major concern (1). While standard viral load measurements and pre-transplant serological status provide critical information, they are still insufficient for evaluating patients at risk of infections (2, 3). Consequently, cell-mediated immune response (CMI) is increasingly studied and appears to be a new biomarker to help clinicians in the management of patients on immunosuppressive drugs. CMI assays measure activated T lymphocytes after *ex vivo* stimulation with specific antigens. These activated T lymphocytes are identified by their ability to express or secrete cytokines, most commonly interferon gamma (IFN-γ), which plays a critical role in controlling viral infection (4). CMI assays have gained increasing attention in clinical practice and could become a key component in international recommendations for the follow-up of SOT patients (5, 6). Some authors have assessed the utility of monitoring CMV-specific T-cell immunity before or after transplantation to predict the risk of CMV infection (7, 8); to guide the duration of antiviral prophylaxis in CMV-seropositive SOT recipients (9–11); to guide treatment decisions at the onset of asymptomatic CMV reactivation (3, 12); and to determine the duration of antiviral treatment and the need for secondary antiviral prophylaxis in CMV syndrome (13, 14).

Several CMI assays are available from commercial and research laboratories, including tests that quantify the IFN-γ secreted in plasma after T-cell activation in whole blood (QuantiFERON CMV and VIDAS-IGRA RUO), tests that determine the number of cells able to secrete IFN-γ after activation (enzyme-linked immunosorbent spot assay, ELISpot-T IFN-γ), or tests based on intracellular cytokine detection after staining by flow cytometry. The ELISpot-T IFN-γ assay allows the detection of virus-specific T-cells by measuring their secretion of IFN-γ in response to viral antigens at the single-cell level. Although it is recognized as the gold standard in CMI exploration, its complexity, high hands-on time, and high variability between laboratories limit its use to specialized centers, creating a need for simpler solutions.

The VIDAS-IGRA RUO (bioMérieux, Marcy-l'Étoile, France) was recently developed, offering an automated and high-throughput alternative. An additional advantage of this assay is the stimulation of T cells with viral antigens directly in whole blood without the need for cell isolation. However, there is a lack of standardization across different CMV-CMI assays. The variability in performance among the CMV-CMI assays may be related to differences in their methods, technologies, antigenic stimulants, and reporting parameters.

In this observational study, as a proof of concept for the VIDAS-IGRA RUO assay, we aim to compare the responses obtained with the VIDAS-IGRA assay and our in-house ELISpot-T IFN-γ assay after stimulation by specific CMV, EBV, and BKV peptides in immunocompromised kidney transplant recipients.

## RESULTS

### Characteristics of ELISpot and VIDAS-IGRA assays

While both the VIDAS-IGRA assay and ELISpot-T assay aim to measure CMI against specific viral antigens, they differ significantly in their design, execution, and interpretation. The major difference lies in the sample matrix. Although both assays require blood collection from patients, the VIDAS-IGRA assay uses unmodified/untransformed whole blood, whereas the ELISpot requires the isolation of peripheral blood mononuclear cells (PBMCs) using a Ficoll-Paque density gradient, which is time-intensive. Moreover, the VIDAS-IGRA can be fully automated using the VIDAS-3 system, while the ELISpot assay is entirely manual, requiring 4 h of trained personnel. Considering the technical time involved, the ELISpot assay is more expensive than VIDAS-IGRA. In our study, frozen PBMCs were used for ELISpot assays. The number of cells and viability after thawing limited the ability to obtain all the results in ELISpot (14 results were missing, 12.5%). However, the ELISpot assay offers quantitative results at the single-cell level (Table 1).

**TABLE 1** Main characteristics of ELISpot and VIDAS-IGRA assays

| | VIDAS-IGRA | ELISpot |
|---|---|---|
| Sample | 3 or 4 mL of whole blood | 250,000 PBMC/well (or 350,000 PBMC/well for BKV)— duplicates |
| Stimulated cells | CD4 and CD8 T lymphocytes | CD4 and CD8 T lymphocytes |
| Results | IFN-γ quantification in supernatant (UI/mL) | Spot quantification: number of PBMCs secreting IFN-γ (spot-forming unit/$10^6$ PBMC) |
| Antigen-activating T cells | Cytomegalovirus lysate (bioMérieux) or customer overlapping peptide pool | Customer overlapping peptide pool |
| Advantages | Sensitive<br>Reproducible<br>CD4 and CD8 response<br>CMI evaluation against several antigens (CMV, EBV, BKPyV)<br>Automated<br>Standardized | Sensitive<br>Reproducible<br>CD4 and CD8 response<br>CMI evaluation against several antigens (CMV, EBV, BKPyV) on the same plate<br>Quantification of other cytokines (TNF-α and IL-2) in the same well (fluorospots multichannel) is possible<br>Avoid interference with circulant IFN-γ<br>Possible in neutropenic patients |
| Threshold value for a positive test | >0.1 UI/mL after specific stimulation and negative control <0.08 UI/mL | Number of spots >2 times the number of spots in the negative control |
| Predictive cut-off value of protection against opportunistic viral reactivation | Not defined—depends on the context of each study | |
| Technical time | Incubation: 16 h<br>Technical time: 1 h or 5 min in automated mode | Incubation: 24 h<br>Technical time: Ficoll 1 h + 4 h ELISpot |
| Drawbacks | No information on cell phenotype | No information on cell phenotype<br>Need to isolate PBMC (time-consuming, specialist staff)<br>Interpretation by expert staff<br>Non-standardized |

## Evaluation of CMV-specific T-cell response

As represented in Fig. 1, we assessed the proportion of positive and negative cellular immune responses measured by VIDAS-IGRA and ELISpot according to antigen stimulation. For CMV stimulation conditions, ELISpot was considered positive if spots/250,000 PBMC > 20 and if VIDAS-IGRA index > 0.2 UI/mL. For CMV PepMix Human CMV pp65 peptide pool (pp65) and the PepMix Human CMV IE-1 (IE-1) mixed with pp65 peptide pool (IE-1 + pp65), from JPT Peptide technologies (JPT), both assays yielded similar positivity rates, indicating good concordance (79.2% for pp65 and 80% for peptide pools [pp65 + IE-1]). With CMV lysate (bioMérieux), VIDAS-IGRA identified a slightly higher proportion of positive responses than ELISpot (76% versus 64%).

Using ELISpot, 79.2% (19/24) of patients had a positive response after pp65 activation, 80% (20/25) after the peptide pools (pp65 + IE-1) activation, and 64% (16/25)

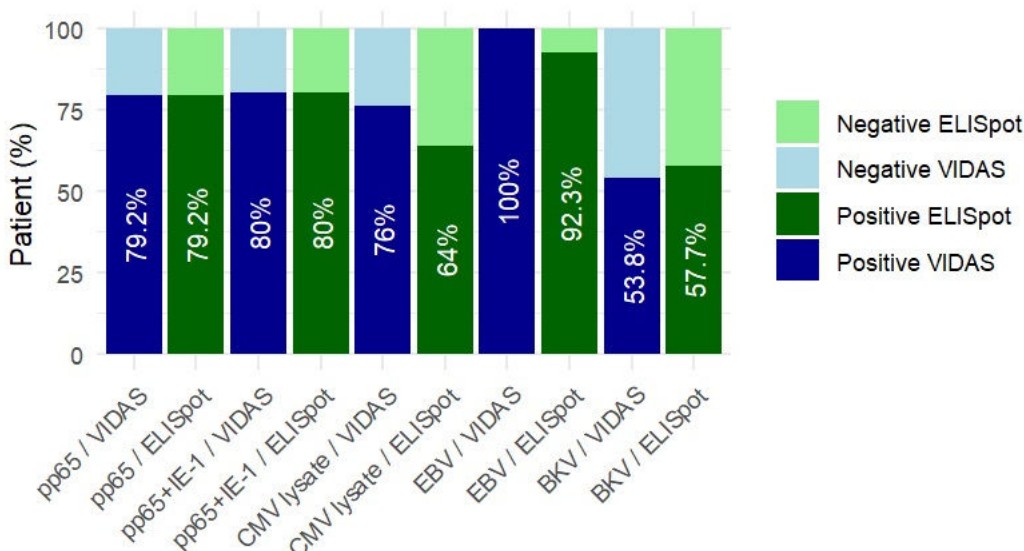

**FIG 1** Qualitative study of cytomegalovirus/Epstein-Barr virus/polyomavirus BK-specific immune response evaluated by ELISpot and VIDAS-IGRA RUO (bioMérieux). For CMV stimulation conditions, ELISpot was considered positive if spots/250,000 PBMC > 20 and if VIDAS-IGRA index > 0.2 UI/mL. For EBV stimulation conditions, ELISpot was considered positive if spots/250,000 PBMC > 10 and if VIDAS-IGRA index > 0.08 UI/mL. For BKV stimulation conditions, ELISpot was considered positive if spots/350,000 PBMC > 10 and if VIDAS-IGRA index > 0.08 UI/mL ($n = 24$ patients with results in both assays for pp65 activation, 25 patients for pp65 + IE-1 and CMV lysate activation, and 26 patients for EBV and BKV).

after lysate activation. In VIDAS-IGRA, 79.2% (19/24) of patients had a positive response after pp65 activation, 80% (20/25) after peptide pool activation, and 76% (21/25) after lysate activation (Table 2). Overall agreement between VIDAS-IGRA and ELISpot ranged from 80% to 100%, depending on antigen stimulation. The highest concordance was observed with the CMV pp65 + IE-1 peptide pools (100%), followed by the pp65 peptide pool (91.7%) and the lysate (80%). The ELISpot/VIDAS-IGRA concordance was very good after pp65 activation (κ score = 0.78) and excellent after pp65 + IE-1 peptide pool activation (κ score = 0.88). Comparison of the bootstrap-dependent kappa coefficients showed no significant difference between the two conditions ($P = 0.66$).

Table 3 shows CMV-specific CMIs of the patient according to their serological status. None of the CMV-seronegative subjects ($n = 5$) showed a detectable response in either assay after peptide pool (pp65 + IE-1) activation. All CMV seronegative patients had

**TABLE 2** Contingency table showing negative and positive results after different peptide activation[a]

| | pp65 peptide pool | | pp65 + IE-1 peptide pools | | CMV lysate | | EBV peptide pool | | BKV peptide pool | |
| --- | --- | --- | --- | --- | --- | --- | --- | --- | --- | --- |
| | Positive VIDAS | Negative VIDAS | Positive VIDAS | Negative VIDAS | Positive VIDAS | Negative VIDAS | Positive VIDAS | Negative VIDAS | Positive VIDAS | Negative VIDAS |
| Positive ELISpot | 18 | 1 | 20 | 0 | 15 | 1 | 24 | 0 | 12 | 3 |
| Negative ELISpot | 1 | 4 | 0 | 5 | 4 | 5 | 2 | 0 | 2 | 9 |
| % of agreement | | 91.7 | | 100 | | 80 | | 92.3 | | 80.7 |
| Fisher's test (*P*-value) | | 0.002 | | <0.001 | | 0.011 | | 1 | | 0.004 |

[a]The percent agreement was noted, and Fisher's test was performed to test the association between the two assays.

**TABLE 3** Qualitative results of ELISpot and VIDAS-IGRA assays against CMV according to the CMV serological status

| Stimulus | VIDAS-IGRA RUO (bioMérieux) results | Positive CMV serology | Negative CMV serology |
|---|---|---|---|
| Peptide pool (pp65 + IE-1) (n = 28) | Positive | 23 | 0 |
| | Negative | 0 | 5 |
| pp65 (n = 28) | Positive | 23 | 0 |
| | Negative | 0 | 5 |
| CMV lysate (n = 28) | Positive | 21 | 0 |
| | Negative | 2 | 5 |
| | ELISpot-T IFN-γ results[a] | | |
| Peptide pools (pp65 + IE-1) (n = 25) | Positive | 20 | 0 |
| | Negative | 0 | 5 |
| pp65 (n = 24) | Positive | 18 | 1 |
| | Negative | 1 | 4 |
| CMV lysate (n = 25) | Positive | 14 | 2 |
| | Negative | 6 | 3 |

[a]Some data are missing due to the lack of PBMC or cell viability <55%. CMI, cell-mediated immune assay.

negative CMV CMI with the VIDAS-IGRA assay. The three patients with discrepant results presented a positive CMV CMI measured with ELISpot after stimulation with the CMV lysate (26 and 125 spots/250,000 PBMCs) or with pp65 (173 spots/250,000 PBMCs). Two patients with positive CMV serology had negative VIDAS-IGRA results. The same two patients also had negative ELISpot results (6 and 12 spots/250,000 PBMCs). With the ELISpot assay, seven patients with positive CMV serology had <20 spots/250,000 PBMCs. Among these discrepancies, six were obtained after activation by the lysate. Among them, four had <1 UI/mL in VIDAS-IGRA.

We analyzed the concordance and the correlation between the ELISpot and VIDAS-IGRA assay results. Table 4 and Table S1 show a more detailed breakdown of the results obtained with these two assays after different peptide pool or lysate activation. With the pp65 + IE-1 peptide pools, results were consistent. In Table S1b, we noticed that two samples had a result with VIDAS-IGRA > 1 UI/mL and a negative response with the ELISpot assay after lysate activation. On the contrary, one sample had between 100 and 200 spots/250,000 PBMCs, even though the response with VIDAS-IGRA was below 0.2 UI/mL. For the latter, the CMV serology was negative.

As VIDAS-IGRA does not take the number of lymphocytes per patient into account, we normalized the results (relative fluorescence value [RFV]) according to the patient's lymphocyte count (G/L), but this approach did not improve the agreement between the two assays (result not shown).

**TABLE 4** Agreement between ELISpot and VIDAS-IGRA semi-quantitative results obtained after activation of T cells with pp65-IE-1 pools

| | Peptide pool (pp65 + IE-1) | | ELISpot (spots/250,000 PBMCs) | | | | | |
|---|---|---|---|---|---|---|---|---|
| | | | −<br><20 | ±<br>20–100 | +<br>100–200 | ++<br>200–500 | +++<br>>500 | |
| VIDAS-IGRA | <0.2 UI/mL | − | 5[a] | | | | | 5 |
| | 0.2–1 UI/mL | ± | | | | | | |
| | 1–8 UI/mL | + | | 1 | 1 | 3 | 1 | 6 |
| | >8 UI/mL and RFV < 10,000 | ++ | | | 2 | 3 | 4 | 9 |
| | >8 UI/mL and RFV > 10,000 | +++ | | | | 2 | 3 | 5 |
| | Total | | 5 | 1 | 3 | 8 | 8 | 25 |

[a]Patients with negative CMV serology. RFV, relative fluorescence value.

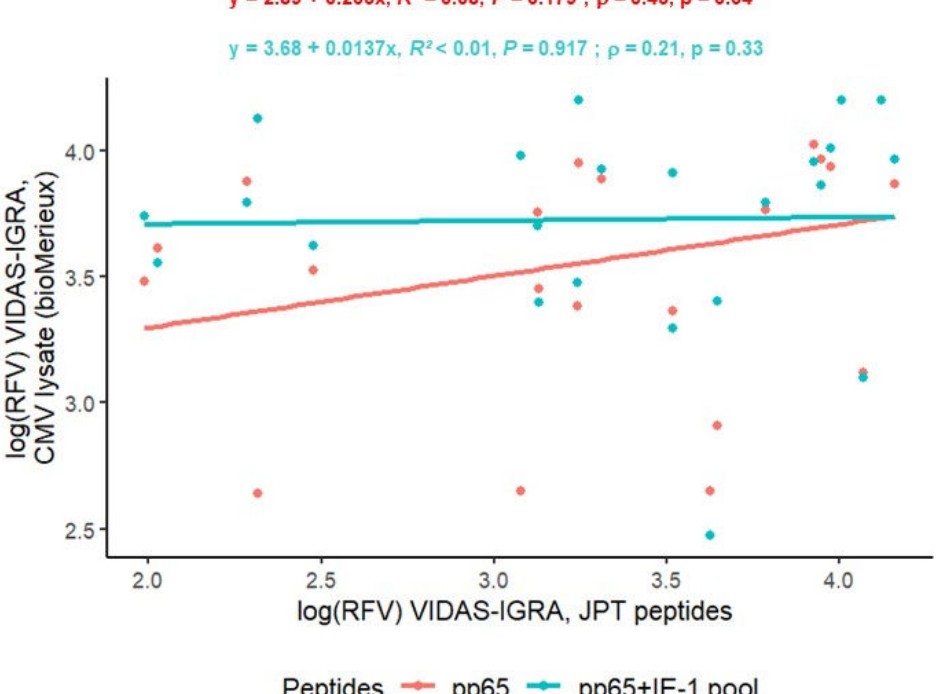

$y = 2.89 + 0.205x$, $R^2 = 0.08$, $P = 0.179$ ; $\rho = 0.43$, $p = 0.04$

$y = 3.68 + 0.0137x$, $R^2 < 0.01$, $P = 0.917$ ; $\rho = 0.21$, $p = 0.33$

**FIG 2** Correlation of positive VIDAS-IGRA positive results (relative fluorescence value > 120) after activation of T cells with CMV lysate (bioMérieux) with other peptides (JPT) (linear regression, $R^2$ ; Spearman's correlation, $\rho$).

We compared the results obtained after T-cell activation with the different stimulating agents for each method individually. Among VIDAS-IGRA-positive results (RFV > 120), quantitative CMI results obtained after activation with CMV lysate or with JPT peptides were moderately correlated for pp65 and low for pp65 + IE-1 pools (Fig. 2, Spearman's correlation, $\rho = 0.43$ for pp65 and 0.21 for the pp65 + IE-1 pools).

Contrarily, a stronger correlation was observed between pp65 and the pp65 + IE-1 pools (linear regression coefficient $R^2 = 0.80$, $P < 0.001$; $\rho = 0.78$, $P < 0.001$, Fig. S2).

Among ELISpot positive results (>20 spots/250,000 PBMCs after each peptide activation, $n = 20$), quantitative CMI results obtained after activation with CMV lysate or JPT peptides were also strongly correlated (Fig. 3, $\rho = 0.7$ for pp65 and 0.67 for the pp65 + IE-1 pools). The linear regression between pp65 and the peptide pools was moderate in ELISpot than in VIDAS-IGRA, but the Spearman's correlation was strong ($R^2 = 0.48$, $P < 0.001$; $\rho = 0.77$, $P < 0.001$, Fig. S3). This means that despite some variability, higher responses measured by one assay were consistently associated with higher responses measured by the other.

Among the 28 transplant patients, only 3 patients (10.7%) had a positive CMV qPCR in whole blood 6 months post-CMI evaluation. CMV qPCR results then became negative without treatment on the next follow-up. All of these patients were CMV-seropositive and showed positive CMI responses in both tests (RFV > 3,000 and > 373 spots/250,000 PBMCs after activation by the peptide pools) (data not shown).

## Evaluation of Epstein-Barr virus-specific T-cell response

All patients tested were EBV-seropositive. For EBV stimulation conditions, ELISpot was considered positive if spots/250,000 PBMC > 10 and if VIDAS-IGRA index > 0.08 UI/mL. As presented in Fig. 1, all samples were positive with VIDAS-IGRA compared to 92.3% with ELISpot. The agreement between the two assays for the measurement of EBV CMI was 92.3% (Table 2). Two patients had negative ELISpot (0 and 7 spots/250,000 PBMCs), while the VIDAS-IGRA results (IFN-γ = 1.17 and 0.27 UI/mL) were positive (Table 5). Among five

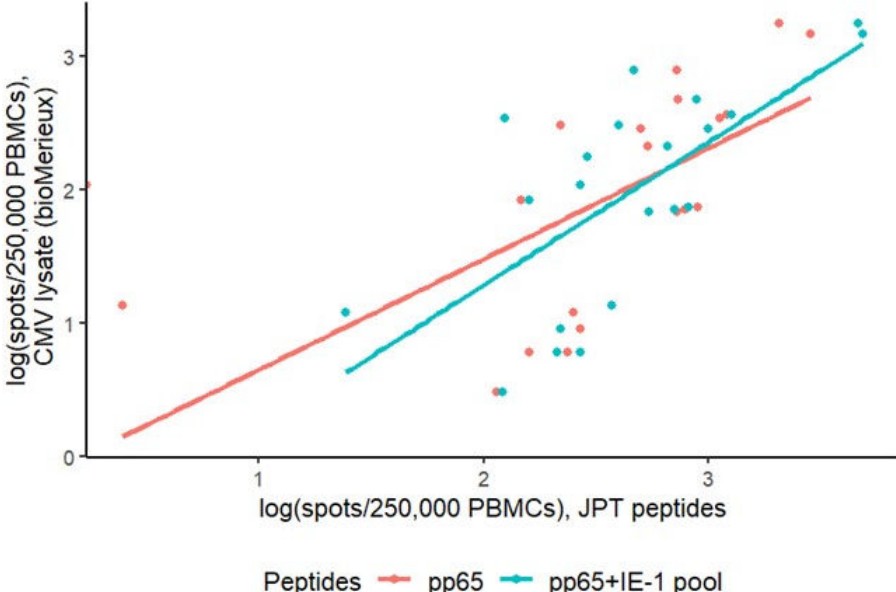

$y = -0.186 + 0.833x$, $R^2 = 0.40$, $P = 0.003$ ; $\rho = 0.7$, $p = 0.00053$

$y = -0.865 + 1.08x$, $R^2 < 0.46$, $P < 0.001$ ; $\rho = 0.67$, $p = 0.00091$

**FIG 3** Correlation of positive ELISpot results (>20 spots/250,000 PBMCs) after activation of T cells with CMV lysate (bioMérieux) or with other peptides (JPT) (linear regression, $R^2$; Spearman's correlation, $\rho$).

patients with an ELISpot below 30 spots/250,000 PBMCs, two had positive EBV qPCR in whole blood (556 and 10,279 copies/mL). The VIDAS-IGRA assay response was >1 UI/mL in only one of them.

Fourteen patients (50%) had at least one whole blood sample with positive EBV qPCR results exceeding 500 copies/mL after blood collection for CMI evaluation. Among them, eight (57%) had CMI response with more than 100 spots/250,000 PBMCs with the ELISpot assay and seven (50%) had >2 UI/mL in VIDAS-IGRA. Comparing patients with or without EBV replication 6 months after CMI measurement, the ELISpot response did not differ (357 spots versus 122 spots, $P = 0.11$), but the VIDAS-IGRA response was significantly higher in patients with EBV replication (RFV = 2,339 versus 1,172, $P = 0.03$).

## Evaluation of polyomavirus BK-specific T-cell response

For BKV stimulation conditions, ELISpot was considered positive if spots/350,000 PBMC > 10 and if VIDAS-IGRA index > 0.08 UI/mL. The agreement between ELISpot and the VIDAS-IGRA assays was at 80.7% (Table 2). Overall, the CMI after BKV peptide pool

**TABLE 5** Agreement between ELISpot and VIDAS-IGRA semi-quantitative results after activation of T cells with EBV peptide pool (JPT)[a]

| | EBV peptide pool | | ELISpot (spots/250,000 PBMCs) | | | | | |
|---|---|---|---|---|---|---|---|---|
| | | | − <10 | ± 10–30 | + 30–100 | ++ 100–350 | +++ >350 | |
| VIDAS-IGRA | <0.08 UI/mL | − | | | | | | |
| | 0.08–0.2 UI/mL | ± | | 1 | | 1 | | 2 |
| | 0.2–2 UI/mL | + | 2 | 1 | 6 | 4 | 1 | 14 |
| | 2–8 UI/mL | ++ | | | 2 | 3 | 1 | 6 |
| | >8 UI/mL | +++ | | 1 | | 2 | 1 | 4 |
| | Total | | 2 | 3 | 8 | 10 | 3 | 26 |

[a]All patients had a positive EBV serology.

activation was weaker than for other viruses (Table 6). Only 53.8% patients had a positive response in VIDAS-IGRA assay versus 57.7% in ELISpot (Fig. 1).

Of the 26 patients, 10 (38%) had at least one positive urine BKV qPCR during the study. Four of them had a positive CMI with ELISpot (>30 spots/250,000 PBMC), and only one had a VIDAS-IGRA > 0.2 UI/mL.

## DISCUSSION

Monitoring of the cellular immune response is increasingly studied and has become a key tool for clinicians managing transplant patients (15, 16). As recent guidelines highlighted, assessing CMI is complex, with various non-standardized assays available. Several meta-analyses have reported differences between assays regarding peptide pools, peptide types, number of PBMCs per well in ELISpot assays, whole blood volume, and activation time (9, 17, 18). Therefore, the comparison and standardization of IGRA assays are essential to evaluate studies and establish detection and prediction thresholds tailored to each clinical situation.

In our study, CMV serological data were consistent with CMV CMI results in 93% of cases. Overall, concordance between VIDAS-IGRA and ELISpot varied depending on the viral antigen used. According to the literature, CMV-specific T cells are not exclusively detectable in CMV-seropositive patients (2, 19). In their cohort, Lùcia et al. (2) reported that approximately 30% of seronegative transplant recipients displayed a detectable anti-CMV CMI response, which was sufficient to confer immune protection and prevent CMV infection. Here, the absence of a CMI response in some seropositive-CMV kidney transplant patients may be attributed to the type of activation, particularly lysate activation, which yielded a lower number of positive CMI responses.

Regarding quantitative CMV-CMI results with different stimulus agents, CMV lysate responses were weakly positively correlated with those obtained with JPT peptides in both assays (Fig. 2 and 3). This result is consistent with the literature suggesting the importance of choosing the right activator peptide pool and comparing results from the same T cell stimuli. For example, Bestard et al. (20) showed that monitoring IE-1-specific T cell responses before transplantation was better than pp65 for predicting the post-transplant risk of CMV infection. Interestingly, our study showed that with both assays, the responses obtained after activation of the pp65 or pp65 + IE-1 pools were well correlated (Fig. S2 and S3).

Head-to-head studies comparing different techniques remain scarce in the literature. Five studies have evaluated and compared anti-CMV CMI monitoring using IGRA assays. Three of them were compared to QuantiFERON-CMV and ELISpot tests in the kidney transplant population. One study highlighted that both assays displayed similar robustness, sensitivity, and specificity and showed that positive results after transplantation were inversely correlated with CMV replication (9). The second study demonstrated better specificity for the ELISpot assay compared to QuantiFERON-CMV. Indeed, a positive ELISpot had a negative predictive value of 94.5% for CMV replication. This better specificity of ELISpot was attributed to the fact that QuantiFERON-CMV measures IFN-γ production following *ex vivo* stimulation with HLA-restricted CMV peptides. Indeed, the ability to choose the peptide pool with ELISpot enables the use of peptide pools

**TABLE 6** Agreement between ELISpot and VIDAS-IGRA semi-quantitative results after T-cell activation by the polyomavirus BK peptide pool

| | BKV peptide pool | | ELISpot (spots/350,000 PBMCs) | | | | | |
|---|---|---|---|---|---|---|---|---|
| | | | − | ± | + | ++ | +++ | |
| | | | <10 | 10–30 | 30–100 | 100–350 | >350 | |
| VIDAS-IGRA | <0.08 UI/mL | − | 9 | | 1 | 2 | 1 | 13 |
| | 0.08–0.2 UI/mL | ± | | 1 | 4 | 1 | | 6 |
| | 0.2–2 UI/mL | + | 2 | | 1 | 3 | 1 | 7 |
| | 2–8 UI/mL | ++ | | | | | | |
| | >8 UI/mL | +++ | | | | | | |
| | Total | | 11 | 1 | 6 | 6 | 2 | 26 |

that are HLA-agnostic (7). Surprisingly, a third study conducted in 53 CMV seropositive transplanted recipients reported that patients with a positive baseline CMI measured by ELISpot result were more likely to develop CMV replication (21). In our study, due to the low number of patients, we did not evaluate the association between CMV replication and our results of CMI assays.

QuantiFERON-CMV was also compared to flow cytometry for intracellular cytokine staining (FC-ICS) enumerating CMV-specific IFN-γ-producing CD8$^+$ T-cells, which performed slightly better in predicting immune protection in CMV seropositive patients (22). A recent Spanish study compared the VIDAS-IGRA and FC-ICS in immunocompetent and transplanted patients and found a slightly higher rate of positive results by FC-ICS due to indeterminate VIDAS-IGRA results (15%). In total, they concluded that there were differences in qualitative and quantitative performances across comparative assays, but the correlation was higher in hematological patients. They assume that this difference could be partially explained by the different timing of sampling across the study and the use of different activating peptides for FC-ICS and VIDAS-IGRA assays (23). Contrary to our study, some positive control VIDAS-IGRA results were indeterminate (positive control < 0.08 UI/mL). Further testing in neutropenic patients could reveal whether VIDAS-IGRA and ELISpot assays are informative, unlike QuantiFERON tests (8). Unlike prior comparative studies, a strength of our study lies in the use of the same activating peptide pool in both assays, and it is the first to compare the CMI assay assessing anti-EBV and anti-BKV CMI. Our study provides a comparison of VIDAS-IGRA and ELISpot-T IFN-γ, focusing on technical and analytical aspects. It is important to note that we had 1 µg of peptide per well for VIDAS-IGRA activation, but subsequent studies have shown that it is possible to reduce to 0.3 µg per well for CMV and EBV activation, thereby reducing the cost of the test (23). We agree with Giménez et al. (23) that the VIDAS-IGRA assay is simple to use, automatable, and suitable for inter-laboratory harmonization. Unlike other IGRA tests, the VIDAS-IGRA RUO is an open assay. For versatility, the proposed stimulus can be used, but users can also introduce their own. Like in-house ELISpots, the VIDAS-IGRA assay accommodates an overlapping peptide pool, not just HLA-restricted CMV peptides as in QuantiFERON-CMV (Qiagen) (7, 21). Focusing on the robustness, ELISpots and VIDAS-IGRA assays showed good qualitative agreement in CMV-CMI evaluation. Agreements in semi-quantitative results were fair to moderate for CMV, EBV, and BKV-CMI evaluation.

Anti-CMV prophylaxis after kidney transplantation is particularly recommended, especially for high-risk donor/recipient pairs (CMV serology D+/R-) or when patients receive T cell-depleting agents. Evaluating CMV CMI in CMV seropositive recipients 1 month after transplantation or after CMV disease is increasingly recommended by scientific societies (16, 24). A positive CMI test enables stopping the CMV prophylaxis earlier. Numerous studies have attempted to define predictive thresholds for the available tests, but no consensus has yet been reached due to the difficulty of interpreting the tests and the lack of standardization (18, 25).

For BKV CMI, a 2021 meta-analysis highlighted the importance of CMI testing in identifying patients at high risk of BKV infection (26). Studies consistently report that the CMI response to BKV is 50–100 times lower than that observed for CMV (17, 27). Schulze Lammer et al. (28) showed that only 30%–40% of BKV-seropositive patients exhibited a detectable anti-BKV CMI, which aligns with our findings. These differences in virus-specific CMI levels may be explained by the varying number of memory T lymphocytes involved in each infection, suggesting that CMI assays targeting different viruses require specific optimizations. In our BKV-specific ELISpot, the number of PBMCs per well was increased, which may have contributed to the poor agreement with results obtained with the VIDAS-IGRA assay, as it does not allow for adjustment of cell count. Optimization of the anti-BKV peptide formulation in the VIDAS-IGRA assay is, therefore, necessary.

To date, very few studies are available to assess anti-EBV CMI and its relevance in predicting post-transplant lymphoproliferative disorder or BKV reactivation (29, 30). The number of spots was often less than 50 spots/1,000,000 PBMCs (30). EBV serological data

were consistent with EBV CMI results measured with ELISpot and VIDAS-IGRA, except for two patients with no ELISpot response and two patients with a very low response in VIDAS-IGRA.

This study has some limitations. The number of subjects included and the late inclusion after transplantation are the main limitations of this study. The lack of data concerning the predictability of both assays should be noted. A larger prospective study of renal transplant recipients, with virus-specific CMI evaluation just after transplantation, followed by monitoring of virus reactivation post-transplantation, will be necessary to determine the predictive ability and cut-off values of these tests for the occurrence of opportunistic virus replication. In our study, the lymphocyte count was >0.5 G/L for each patient. A comparative study on neutropenic patients would be interesting in order to compare the robustness of these two tests. Moreover, discrepancies across the comparison assays were not resolved due to the absence of a reference assay and the lack of external validation of our ELISpot assay.

## Conclusion

This first comparison of VIDAS-IGRA and in-house ELISpot-T IFN-γ assays highlights the need for using the same antigens for T-cell activation in both tests. The VIDAS-IGRA assay showed good agreement with ELISpot for CMV after activation by peptide pools from JPT and EBV CMI, underscoring its robustness and reliability. For BKV, the immune response remains weaker overall, with greater discrepancies between the two assays. A longitudinal study to better investigate the evolution of the response and its link with viral load could be interesting.

The VIDAS-IGRA assay stands out for its simplicity, automation capability, and potential for harmonization across laboratories. Working with whole blood avoids PBMC isolation and limits cell viability loss. Unlike other IGRA assays, the open features of the VIDAS- IGRA allow flexibility in stimulant choice, making it a versatile clinical tool.

## MATERIALS AND METHODS

### Patients and samples

Blood samples from 28 kidney transplant recipients were collected at least 2 years after the transplant at variable times, with a median of 4.2 years (interquartile ratio = 7.7 years) after transplantation. Patients could have had prior viral exposure, but none had antiviral treatment at the time of inclusion. Whole blood was used on the day of collection for the VIDAS-IGRA assay and for the isolation of PBMCs using a Ficoll-Paque density gradient. Isolated PBMCs were cryopreserved in dimethyl sulfoxide and stored in liquid nitrogen for subsequent ELISpot testing. In parallel, opportunistic viral replication in these patients was monitored by retrospectively collecting clinical and biological data from our electronic records, 6 months after the CMI test. The workflow is illustrated in the flowchart (Fig. S1).

### ELISpot-T IFN-γ

Interferon-gamma enzyme-linked immunosorbent spot assays were performed using several activating agents, including an overlapping peptide pool that maps the entire antigen sequences specific to different opportunistic viruses, such as CMV, EBV, and BKV (see below). Additionally, a CMV lysate provided by bioMérieux and recommended in their VIDAS-IGRA RUO assay (bioMérieux, Lyon, France) was tested.

The day before the assay, a 96-well plate was coated with anti-human IFN-γ antibody (1.5 µg per well) and stored at 4°C. On the day of the ELISpot-T IFN-γ, 250,000 viable PBMCs (in 100 µL) were seeded into the wells for CMV and EBV assays. For BKV assays, the number of PBMCs was increased to 350,000 viable cells to enhance test sensitivity. The cell viability threshold was set at 65%. Simultaneously, 0.25 µg of specific activating peptide pool (PepMix, JPT Peptide Technologies GmbH, Berlin, Germany) (in 100 µL) was

added to the corresponding wells. After 24 h of activation at 37°C, PBMCs secreting IFN-γ were detected using biotinylated anti-human IFN-γ antibodies, followed by the addition of streptavidin-alkaline phosphatase conjugate and then the color substrate (BCIP/NBT-plus) (ELISpot Flex: Human IFN-γ [ALP] Kit, Mabtech, Sweden). The resulting spots, representing IFN-γ-secreting cells, were counted using the IRIS 1 plate reader (Mabtech). For each PBMC sample, the specific assay was performed in duplicate wells. A well with RPMI medium alone served as a negative control, and a well containing anti-CD3 antibody (mitogen) served as a positive control. Each result in spots/PBMCs was calculated as the mean value of duplicate measurements after subtracting the response of the negative control well. According to some literature data for anti-CMV assays (7, 31, 32), the positivity thresholds have been defined at 20 spots/250,000 PBMCs for CMV, 10 spots/250,000 PBMCs for EBV, and 10 spots/350,000 PBMCs for BKV.

## VIDAS-IGRA

IGRA assays were performed using the VIDAS 3 system with the VIDAS IGRA RUO kit (bioMérieux). A 200 µL volume of homogenized whole blood was incubated with 200 µL of CMV lysate from the VIDAS STIMM CMV RUO kit (bioMérieux, Lyon, France) or 200 µL of specific peptide pool (PepMix, JPT Peptide Technologies GmbH, Berlin, Germany), equivalent to 1 µg of peptides per well for activation. For each whole blood sample, a negative control (medium = NIL condition) and a positive control (with nonspecific stimulant = MIT condition) were included (VIDAS STIMM BASIC RUO, bioMérieux). Cell activation was performed at 37°C for 16 h. Subsequently, an enzyme-linked immuno-fluorescent assay was carried out on the VIDAS system to quantify the concentrations of IFN-γ secreted in the supernatants (VIDAS IFN-γ RUO, bioMérieux). IFN-γ concentration was calculated after anti-IFN-γ immunocomplex formation on SPR and fluorescence reading. These steps could be fully automated if the laboratory is equipped with a VIDAS 3 system (33). Results are available as UI/mL and fluorescence signal (RFV). The defined thresholds for defining positive results in the VIDAS-IGRA were lowered after EBV and BKV peptide activation (level of IFN-γ < 0.08 UI/mL) compared to after CMV peptide activation (level of IFN-γ < 0.2 UI/mL).

## Antigen-activating T cells

For CMV-specific CMI measurement, an overlapping peptide pool corresponding to the two most immunogenic antigens was used. The first peptide pool corresponded to 138 peptides derived from the pp65 protein (PepMix Human CMV pp65, JPT Peptide Technologies GmbH, Berlin, Germany), the most abundant protein in the CMV tegument. This protein plays a crucial role in preventing the cellular immune response by inducing phosphorylation of early CMV proteins, thereby preventing their recognition by the major histocompatibility complex or by attenuating the interferon response. The second peptide pool corresponded to 120 peptides derived from the immediate-early protein-1 (PepMix Human CMV IE-1, JPT Peptide Technologies GmbH), which modulates the host immune system and stimulates the expression of early viral genes, leading to viral DNA replication. Additionally, the CMV-specific activation was performed using a CMV lysate provided by bioMérieux, derived from CMV cell culture. Both pools cover the entire protein and are HLA-agnostic.

For EBV-specific CMI measurement, a pool of 135 peptides (defined HLA class I and II-restricted T-cell epitopes) derived from 19 immunodominant EBV antigens was used (PepMix Pan-EBV Select, JPT Peptide Technologies GmbH). The proteins primarily represented included immediate-early viral genes (BMLF1, BARF1, and BZLF1) and EBV latent antigens (EBNA1-3 and LMP1-2).

For BKV-specific CMI measurement, a pool of 53 peptides (defined HLA class I and II-restricted T-cell epitopes) derived from 3 immunodominant antigens of polyomavirus BK was used (PepMix Pan-BKVSelect, JPT Peptide Technologies GmbH). The proteins mainly represented included the capsid protein (VP1), large T antigen, and small T antigen (LTA and STA).

To compare the ELISpot and VIDAS-IGRA assays in measuring CMI against opportunistic viruses, identical peptide pools were used in both assays.

## Serological testing

The CMV and EBV serological status was determined in a hospital routine care setting using the Liaison XL instrument (DiaSorin, Saluggia, Italy). This two-step immunoassay quantifies anti-VCA IgM, anti-VCA IgG, anti-EBNA IgG, anti-CMV IgM, and CMV IgG through the use of magnetic particles and a final chemiluminescence detection.

## Viral load quantification

CMV, EBV, and BKV DNA loads were quantified as part of routine hospital care on the day of whole blood collection for CMI measurement, as well as 3 and 6 months thereafter. Analyses were performed using the LightCycler 480 II instrument (Roche, France) with the R-GENE kit (bioMérieux) following DNA extraction with the EMAG platform (bioMérieux) from whole blood samples for all viruses and from urine samples for BKV.

## Statistical analysis

As this was a proof-of-concept study, no sample size calculation was performed. All data are presented as mean ± standard deviation. Qualitative agreement between the two assays was assessed and evaluated using Fisher's test. Inter-method agreement was assessed using Cohen's kappa coefficient. Kappa ($\kappa$) values were interpreted according to the Landis and Koch classification (34). Differences between continuous variables for unpaired comparisons were compared using the Mann-Whitney $U$ test (or linear regression). Correlations were analyzed using Spearman's correlation coefficient ($\rho$). A $P$-value of $< 0.05$ was considered statistically significant. The statistical analysis and visualization were performed using RStudio software (R.4.3.0, Vienna, Austria).

## ACKNOWLEDGMENTS

We thank the bioMérieux company for the donation of VIDAS RUO kits to complete the project and JPT Peptide Technologies GmbH for their help in obtaining the peptide pool. Thank you to Dr. Mohammed Habib and Aurélie Georges for their technical assistance and PBMC isolation in Ficoll-Paque density gradient.

This project received funding from bioMérieux to enable the Ficoll-Paque.

All the authors participated in the research design and in the completion of the research. A.T., R.G., F.B., S.D., and J.L. participated in the writing of the paper and in data analysis. A.T., B.N.-T., and L.F. performed ELISpot assays. A.T. and H.P. performed VIDAS-IGRA analysis.

## AUTHOR AFFILIATIONS

[1]IBS, IRIG, CNRS, CEA, University of Grenoble Alpes, Grenoble, France
[2]R&D ImmunoAssay, bioMérieux, Marcy-l'Étoile, France
[3]Team Epigenetics, Immunity, Metabolism, Cell Signaling and Cancer, Institute for Advanced Biosciences, Inserm U 1209, CNRS UMR 5309, Grenoble-Alpes University Hospital, University of Grenoble Alpes, Grenoble, France
[4]JPT Peptide Technologies GmbH, Berlin, Germany
[5]LifeSciences R&D, bioMérieux, Marcy-l'Étoile, France

## AUTHOR ORCIDs

Aurélie Truffot  http://orcid.org/0000-0002-7991-1733
Julien Lupo  http://orcid.org/0000-0002-6755-3115

## AUTHOR CONTRIBUTIONS

Aurélie Truffot, Methodology, Validation, Visualization, Writing – original draft, Writing – review and editing | Soizic Daniel, Conceptualization, Funding acquisition, Validation | Lucie Fusillier, Data curation, Investigation | Baptiste Nicolas-Truphemus, Resources | Heidi Primard, Formal analysis | Julien Lupo, Writing – review and editing | Thomas Jouve, Supervision | Gregory Hansen, Visualization | Marie Deroussent, Conceptualization, Funding acquisition | Franck Berthier, Funding acquisition, Supervision, Validation, Writing – review and editing | Raphaële Germi, Supervision, Validation

## DATA AVAILABILITY

The data sets generated and/or analyzed in the current study are available from the corresponding author upon reasonable request.

## ETHICS APPROVAL

The University Hospital review board approved this prospective monocentric study (registration RnIPH 2023, protocol Belaswitch; CNIL number: 2205066 v0, MTA: bioMérieux: No. IPM 21254).

## ADDITIONAL FILES

The following material is available online.

### Supplemental Material

**Supplemental material (Spectrum02289-25-S0001.docx).** Fig. S1 to S3; Table S1.

### Open Peer Review

**PEER REVIEW HISTORY (review-history.pdf).** An accounting of the reviewer comments and feedback.

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
