## [Reviewer comments · Microbiology Spectrum]

Microbiology Spectrum

Comparison of two IGRA assays exploring cell-mediated immunity against CMV, BKV and EBV in kidney transplant patients.

Aurélie Truffot, Soizic Daniel, Lucie Fusillier, Baptiste Nicolas-Truphemus, Heidi PRIMARD, Julien LUPO, Thomas Jouve, Gregory Hansen, Marie Deroussent, Franck Berthier, and Raphaële Germi

Corresponding Author(s): Aurélie Truffot, Centre Hospitalier Universitaire Grenoble Alpes Institut de Biologie et de Pathologie

Review Timeline:

Submission Date:	July 28, 2025
Editorial Decision:	October 24, 2025
Revision Received:	January 9, 2026
Accepted:	January 23, 2026

Editor: Bonnie Prokesch

Reviewer(s): The reviewers have opted to remain anonymous.

Transaction Report:

DOI: <https://doi.org/10.1128/spectrum.02289-25>

Re: Spectrum02289-25 (**Comparison of two IGRA assays exploring cell-mediated immunity against CMV, BKV and EBV in kidney transplant patients.**)

Dear Miss Aurélie Truffot:

Thank you for the privilege of reviewing your work. Below you will find my comments, instructions from the Spectrum editorial office, and the reviewer comments.

Revision Guidelines

Sincerely,
Bonnie Prokesch
Editor
Microbiology Spectrum

Reviewer #1 (Comments for the Author):

This manuscript from Truffot, et al demonstrates the comparability of the Vidas IGRA and the ELISpot for detecting T-cell responses in post-transplant patients for CMV, EBV, and BK Virus. This type of testing is in need of further development as well as a more robust understanding of how to employ this testing in this patient population. A few observations/recommendations for consideration are listed below.

1. Clarity in the specific goals of this study should be highlighted. As a proof of concept for the Vidas, the smaller data set across three viral agents is a bit more understandable. The conclusion that the Vidas is robust and reliable is concerning for this reader since there are still large unknowns of how to optimize the assays, how they apply across a wider range of patients, and so on. It seems like more development is needed to consider this to be a robust option for patient monitoring. I also believe that the advantages, etc in Table 1 for a new assay that still requires refinement is a bit premature.
2. The results section could use some additional modifications to make the data sets easier to understand. As an example, line 107-110 states that performance of the two assays was similar then specifies positive response with the ELISpot but no mention of the % positive with the Vidas. It was difficult to understand the data without relying only on the figures.
3. Related to that there was a lot of wording "similar" or "moderate" on line 128 that I am not sure how to interpret. My understanding is moderate per Kappa is >0.4 and the stated result was 0.31. Please consider revising.
4. The importance mentions that all of the patients were treated with belcept. This is not mentioned anywhere else. Is this a concern for the IGRA?

Reviewer #2 (Comments for the Author):

The authors present a qualitative assessment of two assays measuring cell mediated immunity against three DNA viruses important in transplant management and potential complications. The dataset was acquired soundly, but presentation and analysis require increased rigor, especially with regard to statistics.

Major points to consider:

1. Cohen's kappa test for interrater reliability between ELISpot and VIDAS appears incorrectly applied and interpreted. Even in a qualitative study, rigorous application of statistics should be achieved. Two issues raised questions. First, the test may not be appropriate for this small sample size as noted in McHugh 2012, a frequently cited methods paper. Second, interpretation of kappa here appears not aligned with field standards. A value of 0.31 is viewed here as "moderate;" even a value of 0.17 was viewed as "moderate." The paper cited by the authors themselves rates these values as "fair" and "slight." Many would view kappa values this low as evidence of lack of agreement. Application of the test should either be modified or removed. For a qualitative study, it may be best to simply just state the % agreement value between the two assays. This value should be reported for all three viruses and any permutations of stimulating agents used. The application of questionable statistics in an attempt to give quantitative support may weaken rather than strengthen the authors' argument.
2. The conclusion regarding the applicability of the VIDAS-IGRA to BKV is not clear. The abstract notes that "assay agreement was acceptable," but a seemingly opposite evaluation is given in the Results stating that "agreement between ELISpot and the VIDAS-IGRA assays was low." What opinion do the authors want to communicate? The conclusion should be reasoned to a reader in more detail. The kappa scores for the comparison for EBV and BKV are almost the same, 0.22 and 0.21 respectively. Why then is agreement for BKV evaluated as "low" but for EBV evaluated as "fair"?
3. The data in Tables 2 and 3 do not match. Why are there only 25 data points in Table 3 instead of 28 or 26? What happened to the 1 negative VIDAS and 1 negative ELISpot data points from the samples with positive CMV serology?
4. Can Figure 1 be expanded to include EBV and BKV results as well? The graphical representation may help a reader evaluate agreement more effectively than the tables. Here is also a good place to show all % agreement values. What's important is not just that the % positive and negative are similar in both assays, but that the positive and negative results should agree between assays. Displaying the 2x2 contingency tables could achieve that goal even better than the current figure.

Minor points to consider:

1. Chi-squared tests are not appropriate for measuring concordance between the two assays because the sample size is too small. A Fisher's test is technically better with this data.
2. Some interpretation of Spearman's correlations are flawed. "Weakly positively correlated" does not apply to all the values in Figures 2 and 3, which range from 0.21 to 0.7.

Response to reviewer : Comparison of two IGRA assays exploring cell-mediated immunity against CMV, BKV and EBV in kidney transplant patients.

30 December 2025

Dear reviewers,

Thank you for your review and your pertinent comments. We have answered your questions and modified the article to clarify the objective and compare the responses of these two tests to all viral stimuli.

Reviewer #1 (Comments for the Author): This manuscript from Truffot, et al demonstrates the comparability of the Vidas IGRA and the ELISpot for detecting T-cell responses in post-transplant patients for CMV, EBV, and BK Virus. This type of testing is in need of further development as well as a more robust understanding of how to employ this testing in this patient population. A few observations/recommendations for consideration are listed below.

1. Clarity in the specific goals of this study should be highlighted. As a proof of concept for the Vidas, the smaller data set across three viral agents is a bit more understandable. The conclusion that the Vidas is robust and reliable is concerning for this reader since there are still large unknowns of how to optimize the assays, how they apply across a wider range of patients, and so on. It seems like more development is needed to consider this to be a robust option for patient monitoring. I also believe that the advantages, etc in Table 1 for a new assay that still requires refinement is a bit premature.

Response: Thank you for your comment. We have emphasized the objective of this work in order to make it clearer (line 91-94). With regard to the advantages of VIDAS-IGRA compared to ELISpot, this technique remains simpler and faster from an analytical standpoint. The lack of data concerning the predictability of this test should be noted, but this point also concerns the ELISpot technique, which is still underdeveloped in the field of transplant patient monitoring (line 308).

2. The results section could use some additional modifications to make the data sets easier to understand. As an example, line 107-110 states that performance of the two assays was similar then specifies positive response with the ELISpot but no mention of the % positive with the Vidas. It was difficult to understand the data without relying only

on

the

figures.

Response: Thank you for your remark. The result part was modified and results are more explained (line 121-143).

3. Related to that there was a lot of wording "similar" or "moderate" on line 128 that I am not sure how to interpret. My understanding is moderate per Kappa is >0.4 and the stated result was 0.31. Please consider revising.

Response: Your comment has been taken into account. This Kappa test was not appropriate for this small sample size as noted in McHugh 2012. Kappa test was used only to compare assays concordance (line 130) and the interpretation was modified.

4. The importance mentions that all of the patients were treated with belatacept. This is not mentioned anywhere else. Is this a concern for the IGRA?

Response: Thank you for your comment. We agree with your statement and removed the information about the treatment. It's not important for the analysis.

Reviewer #2 (Comments for the Author): The authors present a qualitative assessment of two assays measuring cell mediated immunity against three DNA viruses important in transplant management and potential complications. The dataset was acquired soundly, but presentation and analysis require increased rigor, especially with regard to statistics.

Major points to consider:

1. Cohen's kappa test for interrater reliability between ELISpot and VIDAS appears incorrectly applied and interpreted. Even in a qualitative study, rigorous application of statistics should be achieved. Two issues raised questions. First, the test may not be appropriate for this small sample size as noted in McHugh 2012, a frequently cited methods paper. Second, interpretation of kappa here appears not aligned with field standards. A value of 0.31 is viewed here as "moderate;" even a value of 0.17 was viewed as "moderate." The paper cited by the authors themselves rates these values as "fair" and "slight." Many would view kappa values this low as evidence of lack of agreement. Application of the test should either be modified or removed. For a qualitative study, it may be best to simply just state the % agreement value between the two assays. This value should be reported for all three viruses and any permutations of stimulating agents used. The application

of questionable statistics in an attempt to give quantitative support may weaken rather than strengthen the authors' argument.

Response: Thank you for your remark. In order to make this analysis on a small sample size clearer, we chose to remove the statistical analyses using the Kappa score test (which is not very appropriate here) and added the % agreement on the two tests with all viruses and all peptides tested (Table 2).

2. The conclusion regarding the applicability of the VIDAS-IGRA to BKV is not clear. The abstract notes that "assay agreement was acceptable," but a seemingly opposite evaluation is given in the Results stating that "agreement between ELISpot and the VIDAS-IGRA assays was low." What opinion do the authors want to communicate? The conclusion should be reasoned to a reader in more detail. The kappa scores for the comparison for EBV and BKV are almost the same, 0.22 and 0.21 respectively. Why then is agreement for BKV evaluated as "low" but for EBV evaluated as "fair"?

Response: We understand your comments and have modified the conclusion regarding this comparison in order to convey a clearer message (line 320).

3. The data in Tables 2 and 3 do not match. Why are there only 25 data points in Table 3 instead of 28 or 26? What happened to the 1 negative VIDAS and 1 negative ELISpot data points from the samples with positive CMV serology?

Response: In the all study, 28 patients were included. We had VIDAS-IGRA results for all patients. For ELISpot, some data are missing due to not enough cells or a low viability cell. This explains why there are not always 28 data points in the comparison between the two assays. The number of analyzed samples for each condition has been added to the Table 3.

4. Can Figure 1 be expanded to include EBV and BKV results as well? The graphical representation may help a reader evaluate agreement more effectively than the tables. Here is also a good place to show all % agreement values. What's important is not just that the % positive and negative are similar in both assays, but that the positive and negative results should agree between assays. Displaying the 2x2 contingency tables could achieve that goal even better than the current figure.

Response: Thank you for your relevant comment that really improves the Figure 1. We have added analysis of all viruses and a table showing the percentage of concordance has been added (Table 2).

Minor points to consider:

1. Chi-squared tests are not appropriate for measuring concordance between the two assays because the sample size is too small. A Fisher's test is technically better with this data.

Response: Thank you for your remark. The Chi-squared test was modified for the Fisher's test (Table 2).

2. Some interpretation of Spearman's correlations are flawed. "Weakly positively correlated" does not apply to all the values in Figures 2 and 3, which range from 0.21 to 0.7.

Response: Spearman's correlations results were modified.

Re: Spectrum02289-25R1 (**Comparison of two IGRA assays exploring cell-mediated immunity against CMV, BKV and EBV in kidney transplant patients.**)

Dear Miss Aurélie Truffot:

Your manuscript has been accepted, and I am forwarding it to the ASM production staff for publication. Your paper will first be checked to make sure all elements meet the technical requirements. ASM staff will contact you if anything needs to be revised before copyediting and production can begin. Otherwise, you will be notified when your proofs are ready to be viewed.

Please note that there is a typo graphical error in the new Figure 1 (the EBV columns are incorrect). This can be fixed in the proof stage but should not be missed!

Sincerely,
Bonnie Prokesch
Editor
Microbiology Spectrum

Reviewer #2 (Comments for the Author):

The improvements to the statistical analysis significantly improve the rigor of the manuscript. The experiments are methodologically sound.

Please note that there is a typo graphical error in the new Figure 1 (the EBV columns are incorrect). This can be fixed in the proof stage but should not be missed!